# An Fc-silent OspA monoclonal antibody passively protects mice from tick and intradermal *Borrelia burgdorferi* challenge

Daniel Palmer[1], Atieh Shemshadian[2], Katherine Berman[1], Ariana Nobles[2], Graham G. Willsey[1], Carol Lyn Piazza[1], Grace Freeman-Gallant[1], Michael J. Rudolph[3], Jeff Bourgeois[4], Linden Hu[4], David J. Vance[1,2], Nicholas J. Mantis[1,2]*

1 Wadsworth Center, New York Department of Health, Division of Infectious Disease, Albany, New York, United States of America, 2 Department of Biomedical Sciences, University of Albany, Albany, New York, United States of America, 3 New York Structural Biology Center, New York, New York, United States of America, 4 Deparrment of Microbiology, Tufts University, Boston, Massachusetts, United States of America

☯ These authors contributed equally to this work.
* nicholas.mantis@health.ny.gov

## Abstract

The monoclonal antibody, LA-2, has played a pivotal role in the development of Outer surface protein A (OspA)-based vaccines for Lyme disease, a multisystem illness caused by the tick-borne spirochete, *Borrelia burgdorferi* sensu lato. Of particular significance was the demonstration more than three decades ago that LA-2 equivalent antibody titers, defined by a competitive-inhibition ELISA, serve as a reliable correlate of vaccine-induced protection across different species, including humans. *In vitro* characterization of LA-2 has identified both complement-dependent and -independent activities, although which of these attributes contribute to protection against *B. burgdorferi* remains unresolved. To address this issue, we generated and characterized an "Fc-silent" version of LA-2 IgG1 carrying so-called LALAPG substitutions (L234A, L235A, P329G). We demonstrate that LA-2 LALAPG retained OspA binding activity but was severely attenuated in *in vitro* complement deposition and complement-dependent borreliacidal assays. Nonetheless, LA-2 LALAPG was as effective as LA-2 at passively protecting C3H mice against nymphal tick-mediated *B. burgdorferi* sensu stricto (s.s.) B31 challenge. LA-2 LALAPG was also equivalent to LA-2 in passively protecting BALB/c mice against intradermal *B. burgdorferi* s.s. B31 challenge. In the intradermal challenge model, viable spirochetes were not recoverable 24 h after injection from skin biopsies of mice treated with LA-2 or LA-2 LALAPG, and an influx of pro-inflammatory cytokines and chemokines to the injection site was abrogated. Collectively, these results suggest that LA-2's primary mode of action involves direct physical interactions with the spirochete rather than complement-dependent killing. Elucidating these mechanisms may have implications for understanding the mechanistic correlates of OspA-based vaccine-induced immunity in humans.

**Data availability statement:** All relevant data are within the manuscript and its Supporting Information files.

**Funding:** This work was supported by the National Institute of Allergy and Infectious Diseases (NIAID), National Institutes of Health, Department of Health and Human Services, Contract No. 75N93019C00040 (PI/PD Mantis). This content is solely the responsibility of the authors and does not necessarily represent the official views of the NIH. The funders had no role in study design, data collection and analysis, decision to publish, or preparation of the manuscript. There was no additional external funding received for this study.

**Competing interests:** The authors have declared that no competing interests exist.

## Introduction

Over the past five decades, monoclonal antibodies (mAbs) have emerged as extraordinary tools in the identification of protective antigens and epitopes associated with pathogens of interest, leading to novel vaccines for viruses, bacteria and parasites [1–3]. In the case of the Lyme disease spirochete, *Borrelia burgdorferi* sensu lato, multiple groups working in the 1990s generated large collections of mAbs that led to the identification of Outer surface protein A (OspA) as a candidate Lyme disease vaccine antigen [4–15]. OspA is a lipoprotein expressed at high levels by *B. burgdorferi* within the midgut of its arthropod vector, the black legged tick (*Ixodes scapularis*), where it is proposed to function as an adhesin [16]. Structurally, OspA consists of 21 antiparallel β-strands with a single C-terminal α-helix [17,18]. The N-terminus is anchored in the spirochete outer membrane via a lipid moiety, while the C-terminus projects away (~80 Å) from the bacterial surface and is accessible to antibody attack [19,20]. OspA is downregulated during or just after spirochete transmission to a mammalian host [21,22]. As such, antibodies elicited by OspA-based vaccines are proposed to inhibit one or more steps in *B. burgdorferi* tick-mediated transmission, although the specific mechanisms by which this occurs remains to be fully elucidated.

Among the many OspA mAbs characterized to date, LA-2 has played a particularly significant role in our understanding of OspA-mediated immunity. LA-2 was one of the first OspA-specific mAbs shown to passively protect mice from *B. burgdorferi* needle infection [6] and tick-mediated challenge [23]. And, until just a few years ago, LA-2 was the only protective antibody whose epitope on OspA had been resolved at the structural level [18,24,25]. In the context of Lyme disease vaccine development, LA-2 serological antibody "equivalence," as defined by a competitive ELISA, proved to correlate with protection against tick-mediated *B. burgdorferi* infection in OspA-vaccinated mice and dogs [26]. Remarkably, as part of a large randomized OspA vaccine trial, it was determined in a subset of individuals that LA-2 equivalent titers are also important biomarkers of Lyme disease susceptibility in humans, as individuals with confirmed Lyme disease had lower LA-2 equivalence than those who did not [27,28]. LA-2 continues to be used as a benchmark in the development of next generation OspA vaccines [29].

Despite LA-2's central role in Lyme disease vaccine development, the exact mechanism by which LA-2 protects against *B. burgdorferi* infection remains incompletely defined. In fact, the basic question of whether complement is needed for LA-2's protective activity has not been addressed. While LA-2 has potent complement-dependent borreliacidal activity *in vitro* [23,30,31], evidence indicates that complement (human or mouse) is not active in the tick midgut [32]. Several complement-independent activities have been ascribed to LA-2, including effects on spirochete transmigration, that would be expected to impede *B. burgdorferi* from the tick midgut [33,34]. Defining the contribution of complement in LA-2's mechanism of action is important for understanding correlates of OspA-mediated immunity, especially as clinical trials of next generation OspA vaccines are ongoing [29,35,36]. In this report, we generate an "Fc-silent" version of LA-2 IgG1 that is effectively devoid of *in vitro* complement-dependent borreliacidal activity and characterize its activity in mouse models of tick-mediated and intradermal *B. burgdorferi* sensu stricto (s.s.) B31 *challenge*.

## Materials and methods

### Ethics statement

The mouse experiments described in this study were reviewed and approved by the Institutional Animal Care and Use Committees (IACUC) at the Wadsworth Center (protocol 22−459) and Tufts University-Tufts Medical Center (protocol B2024-50). All investigators had relevant training in laboratory animal health and welfare provided by their institutions. The Wadsworth Center and Tufts University-Tufts Medical Center both comply with the Public Health Service Policy on Humane Care and Use of Laboratory Animals and were issued assurance numbers A3183-01 and A4059-01, respectively. Both facilities are fully accredited by the Association for Assessment and Accreditation of Laboratory Animal Care (AAALAC). Obtaining this voluntary accreditation status reflects that these facilities' Animal Care and Use Program meets all standards required by law and goes beyond the standards as it strives to achieve excellence in animal care and use. Animals were euthanized by carbon dioxide asphyxiation followed by cervical dislocation, as recommended by the Office of Laboratory Animal Welfare (OLAW), National Institutes of Health, approximately 3 weeks after *B. burgdorferi* s.s. B31 challenge (by tick or needle) per the IACUC predetermined experimental endpoint. Under these experimental protocols, the mice did not experience any distress or discomfort, and no adverse events (deaths) were noted during the course of this study.

### Recombinant *B. burgdorferi* s.s. B31 proteins

Recombinant OspA, DbpA and OspC type A derived from *B. burgdorferi* s.s. B31 were expressed in *E. coli* as cited in Table 1.

### LA-2 and LA-2 LALAPG IgG1 expression and purification

Codon optimized $V_H$ and $V_L$ DNA sequences of LA-2 (Antibody Registry RRID: AB_2619693) derived from PDB 1FJ1 [18] were custom synthesized by Life Technologies (San Diego, CA) and cloned into TMV and PVX plant expression vectors containing codon-optimized human kappa and human IgG1 constant regions [39]. The resulting plasmids were transformed into *Agrobacterium tumefaciens*. Four-week-old *N. benthamiana* plants were infiltrated with *A. tumefaciens* carrying plasmids for the expression of heavy and light chains of LA-2 LALAPG. Aerial plant parts were harvested after 7 days and extracted and clarified. The LA-2 and LA-2 LALAPG antibodies were purified by Protein A affinity and anion exchange chromatography [40].

### Antibody affinity determinations by biolayer interferometry (BLI)

Affinity determinations were conducted using an Octet RED96e Biolayer Interferometer (Sartorius, Goettingen, Germany) with Data Acquisition 12.0 software. Biotinylated OspA (5 µg/mL) in PBS containing 2% w/v BSA ("buffer") was captured onto Octet SA (streptavidin) biosensors (Sartorius) for 5 min. After equilibration, sensors were immersed in two-fold serial

**Table 1. Recombinant *B. burgdorferi* s.s. B31 proteins used in this study.**

| Antigen | AA | UniProt ID | References |
|---------|-----|-----------|-----------|
| OspA | 18-273 | P0CL66 | [24] |
| OspB | 50–296 | P17739 | unpublished |
| OspC$_A$ | 38-201 | Q07337 | [37] |
| DbpA | 26-188 | O50917 | [38] |
| DbpB | 21–187 | O50917 | unpublished |

Abbreviations; AA, amino acid; UniProt (https://www.uniprot.org/)

dilutions of mAb starting at 100 nM for 10 min. The sensors were then dipped into buffer for 30 min to allow for dissociation. The raw sensor data were loaded into the Data Analysis HT 12.0 software, grouped and fit using a 1:2 bivalent analyte model.

### Flow cytometric analysis of *B. burgdorferi* s.s. B31 surface labeling

*B. burgdorferi* s.s. B31 surface labeling with LA-2 and LA-2 LALAPG was performed essentially as described [31]. LA-2 and LA-2 LALAPG were 2-fold serially diluted in PBS before incubation with viable *B. burgdorferi* s.s. B31. The ricin-specific mAb, PB10, was used as an IgG1 isotype control (10 µg/mL). Alexa Fluor 647-labeled goat anti-human IgG (H+L) (Invitrogen, Carlsbad, CA) was used as a secondary antibody. Samples were analyzed using a BD FACSCalibur (BD Biosciences, Franklin Lakes, NJ). Bacteria were gated on FSC and SSC to exclude debris, and 20,000 events were counted per condition. Agglutination was calculated as the percent of events in the UL+UR+LR quadrants. Data were analyzed using FlowJo v10.10.0 (BD Biosciences).

### Complement-dependent borreliacidal assays

Complement-dependent borreliacidal assays were performed using a recombinant *B. burgdorferi* s.s. B31-5A4 strain carrying an IPTG-inducible *mscarlet-I* reporter (GGW979), as described [31]. Briefly, GGW979 cultures were grown to mid-log phase in BSKII medium supplemented with gentamicin (50 µg/ml) at 32 °C under static conditions. Spirochetes were harvested by low-speed centrifugation and resuspended in phenol red–free BSKII containing gentamicin (50 µg/ml) to a final density of $3 \times 10^7$ spirochetes/ml.

Cell suspensions were then mixed 1:1 with phenol red–free BSKII supplemented with 20% guinea pig complement (Sigma Aldrich, St. Louis, MO) and 20 nM of one of the following mAbs: LA-2, LA-2 LALAPG, 857−2, PB10. PB10, a ricin toxin-specific antibody, was used as an IgG1 isotype control. Reactions were set up in white 96-well assay plates (Co-Star). Following sample addition, reactions contained $1.5 \times 10^6$ spirochetes, 5 nM of antibody and 10% guinea pig complement. Assay plates were then incubated overnight in a water jacketed incubator at 37°C with 5% $CO_2$. The following day, 1 mM IPTG was added to each well to induce *mScarlet-I* expression. Following a 48-h incubation at 37°C with 5% $CO_2$, the MFI was recorded at 569 nm (excitation)/611 nm (emission) using a Spectramax ID3 plate reader (Molecular Biosystems, San Jose, CA). Raw MFI data was then normalized as described [31]. The data presented is the mean and SD of three independent experiments.

### Multiplex immunoassay

To determine whether mice seroconverted against *B. burgdorferi* s.s. B31, a Luminex-based multiplex immunoassay (MIA) was used to measure antibody responses to several *B. burgdorferi* s.s. B31 antigens in harvested mouse serum. For this assay, recombinant *B. burgdorferi* s.s. B31 antigens OspA, OspB, OspC, DbpA, and DbpB (Table 1) were coupled to Magplex-C microspheres (5 µg antigen/ 1x10⁶ microspheres) using a xMap Antibody Coupling Kit as recommended by the manufacturer (Luminex Corporation, Austin, TX). Beads were protected from light and stored at 2−8°C in xMAP AbC Wash Buffer (5x10⁶ microspheres/mL) until use. Mouse serum samples (1:100) and coupled microsphere stocks (1:50) were diluted in assay buffer (1 x PBS, 2% BSA, pH 7.4). The diluted sera (50 µL) and diluted microspheres (50 µL) were combined in black, clear-bottomed, non-binding, chimney 96-well plates (Greiner Bio-One, Monroe, North Carolina) and incubated at room temperature for 1 hr in a tabletop shaker (600 rpm). Plates were placed on a magnetic separator and washed three times using wash buffer (1 x PBS, 2% BSA, 0.02% TWEEN-20, 0.05% Sodium azide, pH 7.4). To detect mouse antibodies against the indicated *B. burgdorferi* s.s. B31 antigens, goat anti-mouse IgG, Human-ads-PE (SouthernBiotech, Birmingham, Alabama) secondary antibody was diluted 1:500 in assay buffer, added (100 µL) to each well, and allowed to incubate at room temperature for 30 min in a tabletop shaker (600 rpm). Alternatively, to detect remaining LA-2 or LA-2 LALAPG, PE labeled goat anti-Human IgG Fc, eBioscience (Invitrogen) secondary antibody was used. Plates

were washed as previously stated. The microspheres were resuspended in 100 µL of wash buffer and placed back on the tabletop shaker (600 rpm) for 5 minutes prior to analysis using a FlexMap 3D (Luminex Corporation). To establish reactivity cutoffs for each antigen, the average median fluorescent intensity (MFI) of buffer-only wells was multiplied by six. MFIs for each mouse serum sample were divided by the antigen-specific reactivity cutoffs yielding an index value. An index value greater than 1 suggests reactivity above background for, and thus seroconversion against, the given antigen.

## Antibody-dependent complement deposition (ADCD) assay

We modified a flow cytometry-based HIV-1 antibody-dependent complement deposition (ADCD) for use with a Luminex instrument and OspA-coupled beads [41]. Magplex-C microspheres coupled with recombinant *B. burgdorferi* s.s. B31 antigens, OspA and $OspC_A$, were diluted (1:50) and mixed 1: 1 (v/v) with primary antibodies, LA-2 and LA-2 LALAPG (10 µg/mL), then seeded into a 96-well plates, covered in foil, and incubated for 1 h at room temperature (RT) with shaking. Plates were washed twice using a plate magnet and 190 µL of wash buffer (PBS, 2% BSA, 0.02% Tween-20, 0.05% sodium azide, pH 7.4). Following the washes, 200 µL of diluted human complement (1:50; Pel-Freez Biologicals, Rogers, AR) were added to each well and incubated for 20 min at RT with shaking. Plates were washed again then phycoerythrin-tagged mouse anti-C3/C3b/iC3b (1:100; BD) and phycoerythrin-tagged goat anti-human IgG Fc (1:500; Invitrogen) were added to their respected wells and incubated for 30 min. The plates were washed a final time before antibody-bead complexes were resuspended in 100 uL of wash buffer and incubated for 1 min while shaking. The plates were analyzed via a FlexMAP 3D instrument (Luminex Corporation) with results presented in median fluorescence intensity (MFI).

## Mouse model of *B. burgdorferi* s.s. B31 challenge by *Ixodes scapularis* nymphs

Animal studies were conducted with approval by the Institutional Animal Care and Use Committees (IACUC) at the Wadsworth Center and Tufts University-Tufts Medical Center. To generate *B. burgdorferi* s.s. B31 infected *Ixodes scapularis* nymphs, C57BL/6J mice were injected subcutaneously with 10^5 cells/mL log growth phase of *B. burgdorferi* s.s. B31. Two to four weeks later, the mice were infested with naive *Ixodes scapularis* larvae, which were allowed to feed to repletion. Replete larvae were harvested and allowed to molt into mature infected nymphs. As controls, we used naive *Ixodes scapularis* nymphs procured from the Oklahoma State University tick rearing facility [42].

For challenge studies, equal numbers of male and female C3H/HeN mice aged ~6 weeks (Charles River Laboratories, Kingston, NY) were acclimated in the Wadsworth Center's vivarium for 1–2 weeks before the start of the experiment. Mice were monitored daily over the course of each experiment. On study day −1, mice were subcutaneously (SC) administered either LA-2, LA-2 LALAPG or an IgG1 isotype control (anti-*Vibrio cholerae* mAb ZAC-3) (120 µg or 30 µg per mouse) diluted in 200 µL of PBS. The following day, infected or naive nymphal ticks (5 per mouse) were placed on a shaved area of the mouse's dorsum. Nymphs were collected from all mice 3–5 days post placement. Mice which had at least one tick that appeared to be at or near repletion at the time of collection were presumed to be successfully challenged. On study day 21, mice were euthanized (as indicated in the Ethics section), and blood was collected via cardiac puncture for serological analysis. To confirm infection, bladders were also harvested for cultivation of spirochetes in 2 mL BSKII cultures treated with rifampicin (50 µg/mL), fosfomycin (20 µg/mL), and amphotericin B (2.5 µg/mL). Infection status was based on seroconversion using the MIA described above, as well as the presence or absence of live spirochetes in bladder cultures using dark-field microscopy, which were assessed weekly for one month.

## Collection of engorged ticks, dissection, and determination of genome equivalents

Replete or near-replete nymphs that had fed on each experimental mouse was dissected to harvest the midgut tissues. DNA was extracted from tick midgut tissues using the E.Z.N.A.® Mollusc & Insect DNA Kit (Omega Bio-tek, Inc., Norcross, GA) and real-time qPCR was performed to determine the *Borrelia* burden. The single-copy *B. burgdorferi* s.s. B31 *flaB*

 

(flagellin) gene was amplified, and *flaB* copy number was standardized to total gDNA in each sample as measured using the Qubit 4 Fluorometer (Invitrogen) to determine normalized spirochete burdens in tick midguts.

## Mouse model of intradermal *B. burgdorferi* s.s. B31 challenge

BALB/c mice were used for all intradermal challenge studies. While BALB/c mice typically develop less severe arthritis than C3H, infection with as few as 200 spirochetes can induce seroconversion and detectable *Borrelia* gDNA in BALB/c mice [43]. Female BALB/c mice aged ~8 weeks (Taconic Biosciences, Germantown, NY) were acclimated in the Wadsworth Center's vivarium for one week before the start of the experiment. On study day −1, mice were injected intraperitoneally (IP) with LA-2 or LA-2 LALAPG (0.1–120 µg/mouse) in 200 µL PBS. The following day (study day 0), mice were challenged with mid-log phase *B. burgdorferi* s.s. B31-5A4 ($1x10^5$ cells) by intradermal (ID) injection. On study day 21, mice were euthanized (as indicated in the Ethics section), and blood was collected via cardiac puncture for serological analysis. Infection status was determined based on seroconversion using the MIA described above.

To assess the effects of LA-2 and LA-2 LALAPG on *B. burgdorferi* s.s. B31 skin dissemination, a mix of male and female BALB/c mice aged ~6 weeks were injected SC with 30 µg of LA-2 or LA-2 LALAPG in 200 µL PBS, or remained untreated (either administered PBS only or administered an IgG1 isotype control (ZAC-3). The results presented in the paper are the combination of two independent experiments. The following day (study day 0), mice were challenged with mid-log phase *B. burgdorferi* s.s. B31-5A4 ($1x10^5$ cells) by ID injection. Groups of mice were euthanized on days 1, 3 and 7, and from all mice ~5 mm skin biopsies were excised from the injection site (IS), ~1 cm away from the IS, and ~3 cm away from the IS. Knee joints and spleens were also harvested and from each mouse to examine dissemination into distal tissues. Skin biopsies and tissues were rinsed in PBS immediately after collection and placed in 2 mL BSKII medium supplemented with rifampicin (50 µg/mL), fosfomycin (20 µg/mL), and amphotericin B (2.5 µg/mL) for cultivation of spirochetes. Cultures were assessed weekly by dark-field microscopy over the course of four weeks for the presence of viable spirochetes.

## Inflammatory cytokine and chemokine analysis in mouse skin biopsies

Groups of male and female BALB/c mice were SC administered 30 µg LA-2 per mouse on day −1 or left untreated. The following day (study day 0), mice were challenged with mid-log phase *B. burgdorferi* s.s. B31 ($1x10^5$ cells) by ID injection. Five days-post infection, mice were euthanized, and a skin biopsy ~1 cm in diameter was collected from the injection site of each mouse and placed in cytokine extraction buffer containing 0.4M NaCl, 0.05% Tween 20, 0.5% bovine serum albumin, 0.1 mM phenylmethylsulphonyl fluoride, and 20 Ki of aprotinin in PBS. The solution containing the biopsy was homogenized at 5 m/s in a bead beater in 1-min increments, with a 1-min cool down between shakes. This was repeated five times, or until the biopsy was fully homogenized. This solution was centrifuged at 13,000 x *g* for 10 min at 4°C, and the supernatant was collected and then prepared for cytometric bead array analysis. Skin homogenates were diluted 1:2 in assay diluent and processed using the BD Biosciences Cytometric Bead Array (CBA) Mouse Inflammation Kit following the manufacturer's instructions. Samples were analyzed using a BD FACSCalibur (BD Biosciences).

## Statistical analysis

Statistical procedures for experiments are described in the figure or table captions. All statistical analysis of data was performed in R, Graphpad Prism 9.0, and Microsoft Excel. In all experiments, p-values <0.05 are considered significant.

## Results

### Recognition of recombinant and native OspA by LA-2 and LA-2 LALAPG

To generate an Fc-silent version of LA-2, codon optimized DNA sequences encoding the variable heavy chain ($V_H$) was cloned in-frame into human IgG1 Fc and IgG1 LALAPG expression vectors. The IgG1 LALAPG derivative carries

three-point mutations (L234A, L235A, P329G) relative to IgG1 that abolishes complement fixation activity and FcγR recognition [44,45]. The LA-2 $V_L$ coding sequence was inserted into a human kappa expression vector. The $V_H$ and $V_L$ plasmids were transformed into *A. tumefaciens* that was then used to infiltrate *N. benthamiana*. Aerial plant parts were harvested after 7 days and extracted and clarified antibodies were purified to homogeneity by Protein A affinity and anion exchange chromatography [40]. By flow cytometry, LA-2 IgG1 and LA-2 LALAPG were equivalent in their ability to recognize native OspA on the surface of viable *B. burgdorferi* s.s. B31, as well as induce agglutination of those cells (Fig 1). Surface labeling was dose-dependent and resulted in a maximum of ~90% total cell labeling with a median fluorescence intensity (MFIs) exceeding 7500 for LA-2 LALAPG. LA-2 and LA-2 LALAPG also recognized recombinant OspA with similar apparent affinities as measured by BLI (S1 Fig).

### LA-2 LALAPG IgG1 lacks complement fixation and borreliacidal activities

Introduction of the LALAPG (L234A, L235A, P329G) mutations into the Fc region of IgG1 is reported to eliminate complement fixation activity [44,45]. To examine this in the case of LA-2, we adopted an antibody-dependent complement deposition (ADCD) assay to OspA (Fig 2A) [41]. Recombinant OspA was covalently coupled to fluorescent microspheres, then probed with LA-2 IgG1 and LA-2 LALAPG in the presence of human complement. Total mAb binding to the beads was determined using PE-labeled anti-human IgG, while complement deposition was measured using a PE-labeled anti-C3 antibody. The results confirmed that LA-2 and LA-2 LALAPG have equivalent capacities to bind OspA (Fig 2B). However, the two mAbs were starkly different in terms of complement fixation activity. LA-2 demonstrated a dose-dependent increase in C3 deposition that peaked at ~2 μg/ml. LA-2 LALAPG, in contrast, was devoid of any activity even at 10 μg/ml (Fig 2C). These results confirmed that LA-2 LALAPG is unable to fix complement via the classical pathway.

To assess the capacity of LA-2 and LA-2 LALAPG to promote complement-dependent borreliacidal activity, we employed a recently developed fluorescence-based *B. burgdorferi* reporter strain GGW979 [31]. GGW979 is a derivative of *B. burgdorferi* s.s. B31 that expresses the red fluorescent protein, mScarlet, under control of an IPTG-inducible promoter [46]. Upon addition of IPTG to the culture medium, viable *B. burgdorferi* express high levels of mScarlet while dead or dying cells express low or negligible levels of mScarlet. In the assay, 5 nM LA-2 IgG elicited borreliacidal activity when cells were grown in media containing 20% exogenous complement (Fig 3). On the other hand, 5 nM LA-2 LALAPG had no measurable complement-dependent borreliacidal activity with the same complement conditions. Collectively, these results demonstrate that the LA-2 LALAPG retains OspA binding activity but lacks complement-fixing activity.

### LA-2 LALAPG protects mice from tick-mediated *B. burgdorferi* s.s. B31 infection

Having established that LA-2 LALAPG is deficient in complement fixation, we next examined the mAb's ability to protect mice from infection in a tick-mediated *B. burgdorferi* s.s. B31 challenge. Groups of C3H/HeN mice were administered 120 or 30 μg of LA-2 or LA-2 LALAPG by subcutaneous injection and challenged the following day with *B. burgdorferi* s.s. B31-infected *Ixodes scapularis* nymphs. Two additional groups of mice received an IgG1 isotype control (ZAC-3) and received either infected or naïve nymphs. On day 21, the mice were euthanized and assessed for *B. burgdorferi* s.s. B31 infection by serology using the MIA described in the Materials and Methods and recovery of viable spirochetes from bladders. For statistical purposes, a mouse was scored as categorically infected if either readout (seroconversion, culture) was positive. By these metrics, all the mice treated with an isotype control and then challenged with *B. burgdorferi* containing nymphs became infected, whereas those challenged with naïve nymphs did not. LA-2 and LA-2 LALAPG were each protective at the 120 μg dose (p<0.01), but only marginally effective at 30 μg dose, relative to mice that received the isotype control (Table 2). Of particular importance, LA-2 and LA-2 LALAPG were statistically indistinguishable in terms of their protective efficacy (p>0.99). These results demonstrate that LA-2 can protect mice from tick-mediated *B. burgdorferi* s.s. B31 infection in the absence of complement fixation.

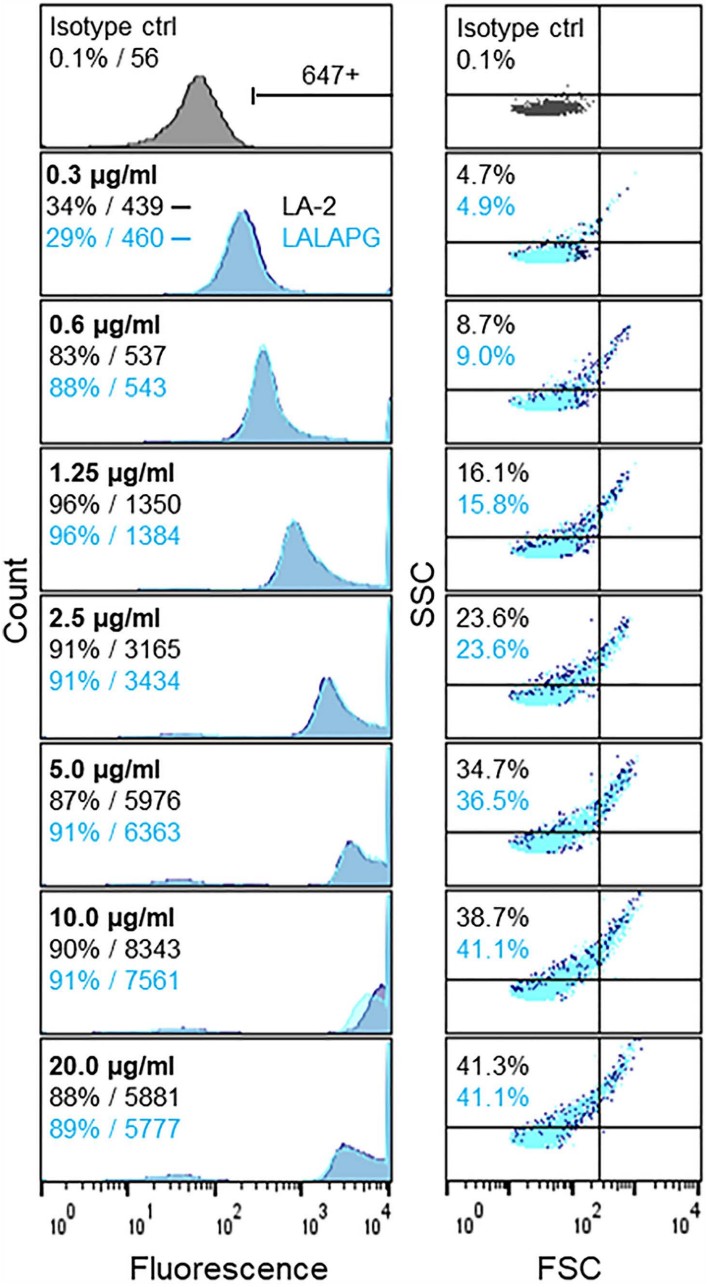

**Fig 1. Reactivity of LA-2 and LA-2 LALAPG with native OspA.** Representative flow cytometry assay of serially diluted LA-2 (dark blue) and LA-2 LALAPG (light blue overlay) reactivity with native OspA on the surface of *B. burgdorferi* s.s. B31. (Left) Fluorescence histogram overlays comparing binding properties. Percent and geometric mean fluorescence intensities of Alexa-647 fluorescently labeled events (under bracket) are indicated. (Right) Forward scatter (FSC) – side scatter (SSC) dot plot overlays comparing agglutination properties. Events increased in FSC and/or SSC (UL, UR, LR quadrants) demonstrate agglutination, and percent of events agglutinated is indicated.

We next examined *B. burgdorferi* loads in ticks that had fed on LA-2 and LA-2 LALAPG treated mice. *B. burgdorferi* (N40 and Westchester strains) burdens in the midgut of ticks that feed on OspA immunized mice were reported to decline or be eliminated entirely within days after engorgement suggesting that OspA antibodies exhibit borreliacidal activities within the

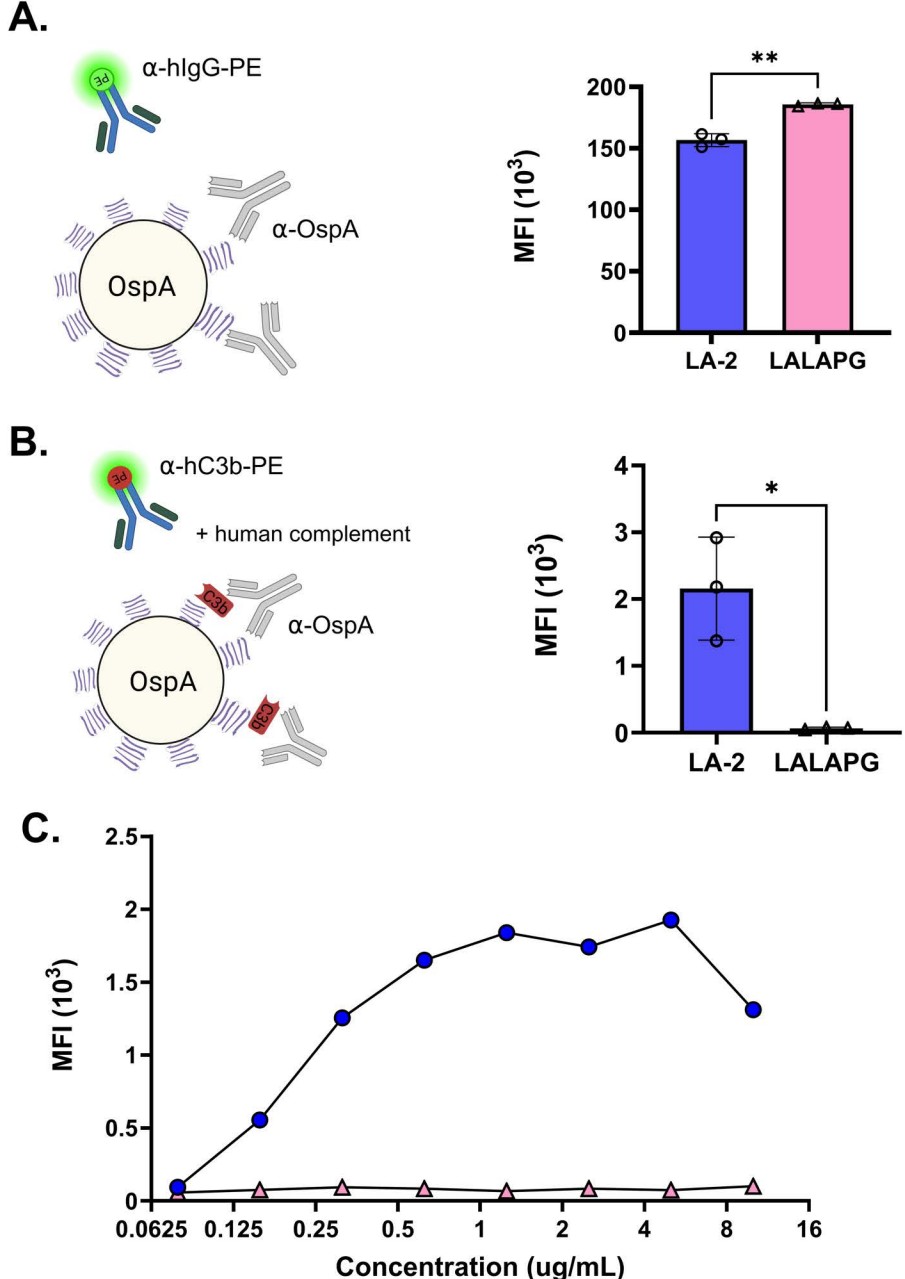

**Fig 2. LA-2 LALAPG is deficient in complement deposition *in vitro*.** (A) Binding (MFI) of LA-2 IgG and LA-2 LALAPG to recombinant OspA. The asterisks indicate a significant difference between groups by Welch's t-test (**P < 0.01). Quantification and comparison of human complement C3 deposition between LA-2 and LA-2 LALAPG in the context of OspA. The asterisk indicates a significant difference between groups by Welch's t-test, where *P < 0.05. **(C)** Dose response of complement C3 deposition of LA-2 and LA-2 LALAPG in the context of OspA.

tick gut [9,32,47,48]. On the other hand, the "LA-2-like" mAb, C3.78, passively protected mice from tick-mediated *B. burgdorferi* s.s. B31 infection without affecting spirochete numbers in the midgut, consistent with a mechanism of action not dependent on borreliacidal activity [49]. To address this issue in the case of LA-2, we collected engorged ticks that had fed on mice treated with LA-2, LA-2 LALAPG- or an isotype control (ZAC-3), dissected the midguts, then quantified spirochete

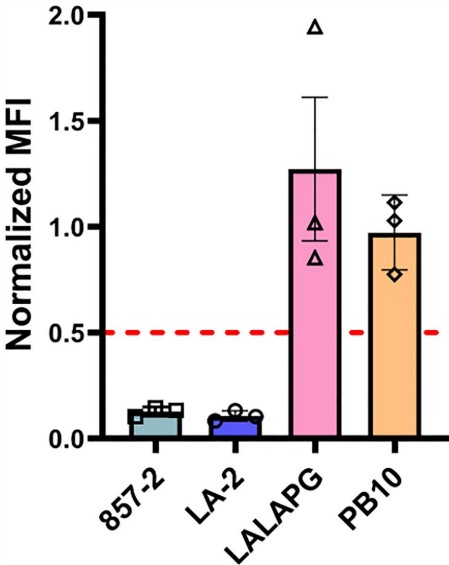

**Fig 3. Complement-dependent borreliacidal activity associated with LA-2 and LA-2 LALAPG.** Mid-log phase *B. burgdorferi* s.s. B31-5A4 carrying an IPTG-inducible *mscarlet-I* reporter (GGW979) were suspended ($1.5 \times 10^6$ cells per reaction) in BSKII medium supplemented with 20% guinea pig complement and 5 nM of the mAbs indicated on the x-axis (857−2, LA-2, LA-2 LALAPG, PB10), as detailed in the Materials and Methods. Following a 48-h incubation, the median fluorescence intensity (MFI; 569 nm excitation/611 nm emission) was determined. The bars are the mean of three independent experiments with each symbol being an independent experiment and the error bars indicating SD. The dashed red line represented 50% killing. The results demonstrate that 857−2 and LA-2 have potent borreliacidal activity as indicated by low normalized MFI, whereas LA-2 LALAPG and the isotype control were devoid of activity.

**Table 2. mAb passive protection in mouse model of tick-mediated B. burgdorferi s.s. B31 challenge.**

| mAb[a] | dose[a] (µg/mouse) | tick[a] | readout (# pos./# total) | | significance (*p*)[c] | |
| | | | serology[b] | culture[b] | vs. IC (+ ticks) | vs. LA-2 |
|---|---|---|---|---|---|---|
| LA-2 | 120 | + | 0/6 | 0/6 | <0.01 | – |
| LA-2 LALAPG | 120 | + | 1/6 | 1/6 | 0.03 | >0.99 |
| LA-2 | 30 | + | 2/5 | 1/5 | 0.18 | – |
| LA-2 LALAPG | 30 | + | 3/6 | 2/6 | 0.18 | >0.99 |
| IC | 30 | + | 5/5 | 5/5 | – | – |
| IC | 30 | – | 0/2 | 0/2 | – | – |

[a], groups of mice were administered LA-2, LA-2 LALAPG, or an isotype control (IC) at 120 or 30 µg/mouse, and challenged with infected (+) or naïve (-) nymphal ticks. [b], number of positive mice/total mice per group. [c], significance (Fisher's exact test with Benjamani-Hochberg procedure for the FDR) was determined using readout with highest infection status. p-values <0.05 are considered significant.

burdens using qPCR. There was no significant reduction in spirochete burdens in the tick midgut in ticks that fed on LA-2 or LA-2 LALAPG-treated mice (S2 Fig), as compared to the isotype control-treated group. Thus, LA-2 or LA-2 LALAPG do not appear to exhibit borreliacidal activity in the context of the tick midgut environment.

## LA-2 and LA-2 LALAPG passively prevent mice from intradermal *B. burgdorferi* s.s. B31 infection

While it is known that LA-2 and other OspA antibodies protect mice from disseminated *B. burgdorferi* infection even when spirochetes are delivered by intradermal injection, there are no reports examining whether spirochete

clearance is dependent on complement [6,50,51]. Considering the sensitivity of *B. burgdorferi* to the classical complement pathway [52], we reasoned that LA-2 would prevent disseminated infection following intradermal challenge, while LA-2 LALAPG would not. It has been reported that OspA expression is nearly uniformly positive on the surface of *B. burgdorferi* s.s. B31 cultured in vitro [53], which we verified through binding to LA-2 and LA-2 LALAPG (Fig 1). Next, groups of BALB/c mice were administered LA-2 or LA-2 LALAPG at 120 µg per mouse then challenged the following day with the same strain of viable *B. burgdorferi* s.s. B31 ($10^5$ cells) by intradermal injection. Three weeks later, mice were euthanized and assessed for seroconversion using the MIA described in the Materials and Methods. All six mice in the LA-2 treated group and five of the six mice in the LA-2 LALAPG group were seronegative, consistent with the animals being resistant to *B. burgdorferi* s.s. B31 challenge (Table 3). From this experiment, we conclude that neither Fc effector functions nor complement fixing activity are required for LA-2's protective activity following intradermal spirochete challenge.

To investigate how LA-2 and LA-2 LALAPG perform at limiting doses, we carried out a pilot study to establish the minimum amount of LA-2 required to protect BALB/c mice against *B. burgdorferi* s.s. B31 intradermal challenge. Those studies indicated that as little as 1 µg of LA-2 IgG per mouse was sufficient to render *B. burgdorferi* s.s. B31 non-infectious (S1 Table). We therefore compared LA-2 and LA-2 LALAPG side by side at doses of 1, 0.3 and 0.1 µg mAb per animal in intradermal *B. burgdorferi* s.s. B31 challenge. At the 1 µg dose, one of the six mice in the LA-2 treatment group was infected at day 21, while two of the six mice in the LA-2 LALAPG group were infected (Table 3). Although neither mAb conferred significant protection at the two lower doses tested (0.3 and 0.1 µg per mouse), LA-2 treated mice fared slightly better than the LA-2 LALAPG treated mice in both cases. We conclude that LA-2 IgG protection in the intradermal challenge model is independent of Fc-mediated activities at high antibody concentrations but possibly important when antibody is limited.

### LA-2 and LA-2 LALAPG clear viable spirochetes from *B. burgdorferi* s.s. B31-infected mouse skin

The fact that both LA-2 and LA-2 LALAPG treatments inhibited systemic *B. burgdorferi* s.s. B31 dissemination in the mouse model of intradermal challenge prompted us to examine early bacterial dissemination kinetics in mice when treated with LA-2 or LA-2 LALAPG. To do this, skin biopsies were collected from groups of 2−4 mice treated with LA-2, LA-2 LALAPG, or untreated control mice. Skin biopsies were collected from three locations on each mouse: the injection site (IS) on the ventral side of the animal, ~1 cm from the IS, and ~3 cm from the IS on the dorsal side of the animal (Fig 4). Knee joints and spleens were also collected from each mouse to examine dissemination into distal tissues. Groups of mice were sacrificed at three time points post-infection: days 1, 3 and 7. Skin biopsies were cultured in BSKII medium for a month and scored regularly for appearance of viable spirochetes. In the control mice, viable spirochetes were recovered from skin on days 1, 3 and 7, the kinetics of which coincided with the reported kinetics of *B. burgdorferi* dissemination [54,55]. Moreover, we also observed that three of the four untreated control mice also had positive knee and spleen cultures. In contrast, skin biopsies from LA-2-treated mice were culture negative

Table 3. mAb passive protection in mouse model of *B. burgdorferi* s.s. B31 ID challenge.

| Dose (µg/mouse) | LA-2 (*p* =)[a] | LA-2 LALAPG (*p* =)[a] | LA-2 vs. LA-2 LALAPG[b] |
|---|---|---|---|
| 120 | 0/6 (<0.01) | 1/6 (0.02) | >0.99 |
| 1 | 1/6 (0.046) | 2/6 (0.09) | >0.99 |
| 0.3 | 3/6 (0.54) | 5/6 (>0.99) | 0.82 |
| 0.1 | 3/6 (0.27) | 6/6 (>0.99) | 0.27 |
| 0 | 6/6 | -- | – |

[a], number of mice that seroconverted/total mice per group with significance compared to the infected control (0 µg/mouse dose) determined by Fisher's exact test with Benjamani-Hochberg procedure for the FDR indicated in parentheses; [b] significance (p values), determined by Fisher's exact test with Benjamani-Hochberg procedure for the FDR, comparing numbers of infected mice between LA-2 and LA-2 LALALPG at each dose indicated.

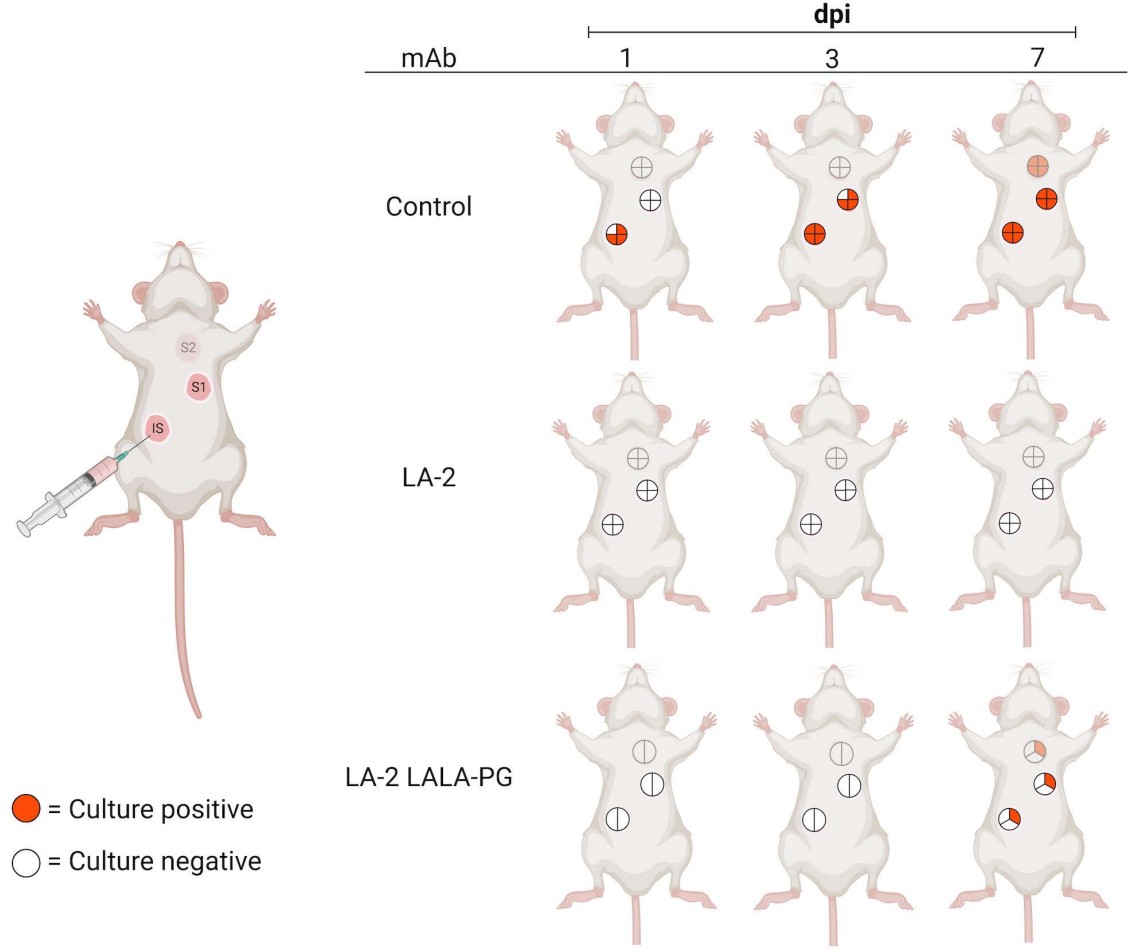

**Fig 4. LA-2 and LA-2 LALAPG prevent spirochete dissemination through skin.** Groups of 2–4 mice were administered LA-2, LA-2 LALAPG, or remained untreated, then intradermally challenged one day later with *B. burgdorferi* s.s. B31. On days 1, 3, and 7 post challenge, these groups of mice were euthanized, and skin biopsies were harvested at the injection site (IS), ~1 cm from the IS (S1), and ~3 cm from the IS (S2) and subsequently cultured. The pie chart subdivisions indicate the number of skin samples (mice) assayed per treatment and time point, with one sample collected from each skin site per mouse. Red subdivisions indicate culture positivity for motile spirochetes, while white subdivisions indicate no viable spirochetes detected in the culture. Shown are the combined results from two independent experiments. Created in BioRender. P, D. (2026) https://BioRender.com/74fgdnw.

at all locations and time points examined (Fig 4). The results were similar for LA-2 LALAPG treatment, with just one of three animals showing positive cultures on day 7 (Fig 4). These results indicate that, in the presence of LA-2 and LA-2 LALAPG, viable *B. burgdorferi* s.s. B31 spirochetes are cleared rapidly at or very near the injection site, thereby arresting dissemination before it even gets started.

The absence of viable spirochetes in the skin biopsies led us to hypothesize that local inflammation may contribute to LA-2-mediated spirochete clearance. To test this, we examined skin biopsy homogenates for the presence of mouse inflammatory chemokines and chemokines TNF-α, IFN-γ, MCP-1, IL-6, IL-10, and IL-12p70. At five days following injection, we observed elevated levels of TNF-α, IFN-γ, IL-6 and especially MCP-1 in untreated mice when compared to uninfected mice (Fig 5). However, in infected mice that were pretreated with LA-2, analytes showed cytokine concentrations similar to levels of uninfected mice (Fig 5). Thus, LA-2 treatment is not associated with residual inflammation in the skin and may clear the spirochetes before a cytokine response can be generated.

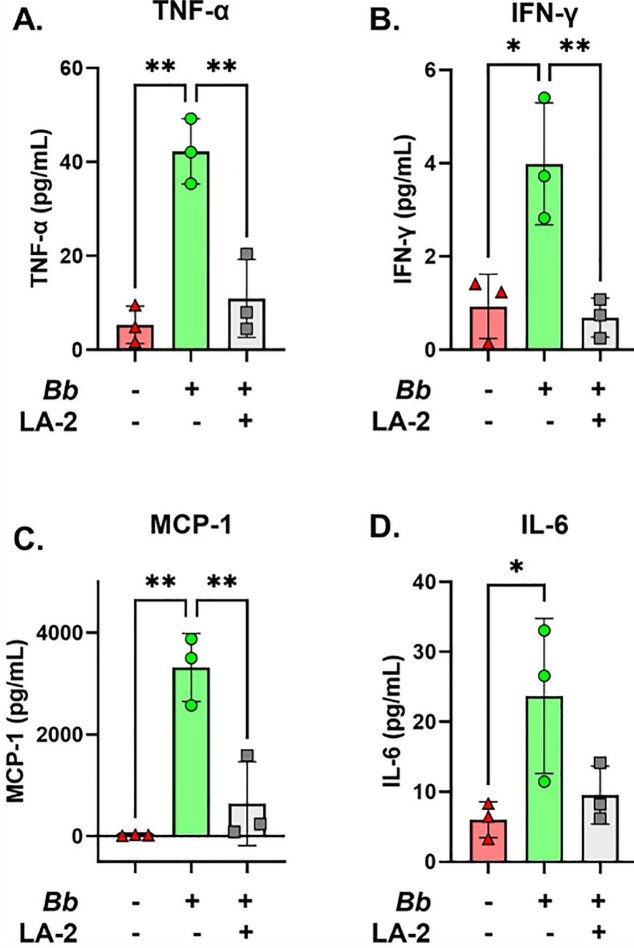

**Fig 5. Cytometric bead array analysis of injection site skin biopsies of BALB/c mice treated with LA-2 and ID infected with *B. burgdorferi* s.s. B31.** On study day –1, mice were SC administered 30 µg/mL of LA-2 or were left untreated. Mice were ID injected with either *B. burgdorferi* s.s. B31 or PBS on day 0 and the injection site skin was harvested on day 5 for CBA analysis. Skin samples were diluted 1:2 in assay diluent for analysis. Statistics performed by one-way ANOVA, no matching or pairing, corrected for multiple comparisons using Tukey's test. 95% confidence interval. *$p < 0.05$, **$p < 0.01$.

## Discussion

In this report, we generated and characterized an "Fc-silent" derivative of LA-2 as a tool to investigate the role of complement in passive protection afforded by LA-2 in both tick- and needle-mediated *B. burgdorferi* s.s. B31 challenge models. The Fc element of LA-2 was rendered silent by the addition of the so-called LALAPG substitutions (L234A, L235A, P329G), a modification that is gaining wide recognition for its research and clinical applications [45,56,57]. We confirmed that LA-2 LALAPG retained OspA binding activity comparable to the parenteral LA-2 IgG1 but was markedly attenuated for *in vitro* complement fixation and complement-dependent borreliacidal activity.

When tested *in vivo*, we found that LA-2 LALAPG was as effective as LA-2 IgG in passively protecting mice from tick-mediated *B. burgdorferi* s.s. B31 challenge, indicating that neither antibody-mediated complement fixation nor complement-dependent borreliacidal activity were necessary to inhibit spirochete infectivity. In this respect, our results agree with Gipson and de Silva who reported that the "LA-2-like" monoclonal antibody C3.78 blocks tick transmission of

*B. burgdorferi* in the absence of host complement [8,49]. Those studies were conducted using complement-deficient (C3) mice and C3.78 Fab fragments. de Silva and colleagues also demonstrated that C3-deficient mice actively immunized with OspA were also protected against *B. burgdorferi* infection, further reinforcing the notion that host complement is not required for transmission blocking activity of OspA vaccines [32].

Furthermore, our results support a model in which LA-2 and LA-2 LALAPG inhibit *B. burgdorferi* s.s. B31 transmission without affecting the number of spirochetes within the tick midgut. This observation is consistent with C3.78's mode of action in which low dose (~60 μg/mouse) antibody protected mice from tick-mediated *B. burgdorferi* s.s. B31 challenge without a concomitant reduction in spirochete numbers in tick tissues [49]. We would attribute this observation to LA-2's inhibitory effects on spirochete migration [34]. Using a two compartment Transwell system, we reported recently that spirochete movement from the lower to upper chambers is reduced by >99% in the presence LA-2 or LA-2 LALAPG. Inhibition of transmigration coincides with LA-2's ability to promote spirochete agglutination, alternations in membrane permeability, and even bleb formation [33,58].

However, other mechanisms may also be at play. Gipson and de Silva, for example, speculated antibodies like C3.78 influence spirochete gene expression patterns, including the requisite *ospA* to *ospC* transition during transmission [23,49]. It is also possible that passive immunization with LA-2 and LA-2 LALAPG antibody do not result in a decline in tick midgut burdens because they do not interfere with OspA binding to tick receptor for OspA (TROSPA) in the tick midgut. TROSPA is highly expressed in flat nymphs and downregulated upon engorgement, serving as an anchor of *B. burgdorferi* s.s. B31 to its vector [16,59]. Whether competition between the two occurs has yet to be elucidated.

While LA-2 has the capacity to interfere with *B. burgdorferi* transmission within the context of the tick, it also reduces infectivity of *B. burgdorferi* within the mouse. Indeed, LA-2 was originally identified as being capable of passively protecting *scid* mice from subcutaneous *B. burgdorferi* challenge [6]. We confirmed and extended that original observation by demonstrating in both BALB/c and C3H mice that remarkably low doses of passively administered LA-2 were sufficient to not only confine but seemingly eliminate *B. burgdorferi* s.s. B31 from the site of intradermal inoculation within hours. LA-2 LALAPG had similar properties, indicating that clearance of *B. burgdorferi* s.s. B31 from the skin environment occurs without complement or Fc effector functions. These observations may be of clinical importance, as they suggest that if *B. burgdorferi* evades immunity within the context of the tick body, any spirochetes that still express OspA upon entry into the skin will encounter a second line of defense [21,60]. While the underlying mechanism by which LA-2 promotes clearance of spirochetes from the skin environment without Fc effector functions is unknown, there are interesting parallels with antibody-mediated clearance of malaria parasites in this same environment that involve motility arrest and membrane shedding [61,62].

In summary, we have demonstrated that LA-2, the well-characterized monoclonal antibody directed against the C-terminus of OspA, functions in both the tick and mammalian environments to limit *B. burgdorferi* infection without the need for Fc effector functions, such as complement fixation and FcγR interactions. It is unclear whether LA-2 equivalence, as defined by a competitive ELISA, which correlates with immunity to Lyme disease in animal models and humans reflects functional activities in vivo or simply a proxy for other activities. Nonetheless, our study makes a case for LA-2's primary mode of action involving direct physical interactions with the spirochete rather than complement-dependent killing. Elucidating these mechanisms may have implications for our understanding of the mechanistic correlates of OspA-based vaccine-induced immunity in humans.

## Supporting information

**S1 Table. LA-2 dose-down in an intradermal *B. burgdorferi* s.s. B31 challenge.**
(PDF)

**S1 Fig. LA-2 and LA-2 LALAPG have similar affinity for OspA.** Biolayer Interferometry sensorgrams for (A) LA-2 and (B) LA-2 LALAPG. Streptavidin coated sensors were dipped in biotinylated OspA, then equilibrated in Ocet buffer (PBS

plus 2% BSA). Sensors were then dipped in a two-fold dilution series of antibody, from 100 down to 1.56 nM. mAb was allowed to bind for 10 minutes, at which point the sensors were dipped into buffer alone and dissociation was measured for 30 minutes. Sensorgrams were analyzed with Octet Analysis Studio and globally fit with the 1:2 bivalent analyte model. (PDF)

**S2 Fig. *B. burgdorferi* B31 burdens in the tick midguts are not significantly reduced after feeding on LA-2 and LA-2 LALAPG treated mice.** Engorged *Borrelia*-infected ticks were harvested after feeding on mice passively immunized with an isotype control, LA-2 or LA-2 LALAPG at 120 or 30 μg/mouse. Ticks were dissected and gDNA extracted from tick midguts for qPCR. Normalized midgut spirochete burden means ± SEMs depicted above and compared across treatment groups using the Kruskal-Wallis test. Shown are comparisons of LA-2 and LA-2 LALA-PG treated groups to ZAC-3 (isotype control) treatment. ns = not significant. (PDF)

## Acknowledgments

We are grateful to Dr. Michael Pauly and colleagues ZabBio for generating LA-2 LALAPG. We thank Drs. Renji Song and Jennifer Yates of the Wadsworth Center's Immunology core for assistance with flow cytometry and the Media and cell culture core for formulating BSK II medium. We thank the Wadsworth Center veterinary staff for assistance with mouse studies. We extend a special thanks to Ms. Elizabeth Cavosie (Wadsworth Center) for administrative assistance.

## Author contributions

**Conceptualization:** Nicholas J. Mantis.

**Formal analysis:** Daniel Palmer, David J. Vance, Nicholas J. Mantis.

**Funding acquisition:** Nicholas J. Mantis.

**Investigation:** Daniel Palmer, Atieh Shemshadian, Katherine Berman, Ariana Nobles, Graham G. Willsey, Carol Lyn Piazza, Grace Freeman-Gallant, David J. Vance, Nicholas J. Mantis.

**Methodology:** Daniel Palmer, Atieh Shemshadian, Katherine Berman, Ariana Nobles, Graham G. Willsey, Carol Lyn Piazza, Grace Freeman-Gallant, David J. Vance, Nicholas J. Mantis.

**Project administration:** Nicholas J. Mantis.

**Resources:** Michael J. Rudolph, Jeff Bourgeois, Linden Hu.

**Supervision:** Nicholas J. Mantis.

**Validation:** Daniel Palmer.

**Writing – original draft:** Daniel Palmer, Atieh Shemshadian, Katherine Berman, Ariana Nobles, Graham G. Willsey, Carol Lyn Piazza, David J. Vance, Nicholas J. Mantis.

**Writing – review & editing:** Daniel Palmer, Graham G. Willsey, David J. Vance, Nicholas J. Mantis.

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
