## [Decision Letter · Decision Letter 0]

28 Jan 2026

Dear Dr. Mantis,

Thank you for submitting your manuscript to PLOS ONE. After careful consideration, we feel that it has merit but does not fully meet PLOS ONE’s publication criteria as it currently stands. Therefore, we invite you to submit a revised version of the manuscript that addresses all of the points raised during the review process.

We look forward to receiving your revised manuscript.

Kind regards,

Brian Stevenson, Ph.D.

Academic Editor

PLOS One

Journal Requirements:

“This work was supported by the National Institute of Allergy and Infectious Diseases (NIAID), National Institutes of Health, Department of Health and Human Services, Contract No. 75N93019C00040 (PI/PD Mantis). This content is solely the responsibility of the authors and does not necessarily represent the official views of the NIH.”

“This work was supported by the National Institute of Allergy and Infectious Diseases (NIAID), National Institutes of Health, Department of Health and Human Services, Contract No. 75N93019C00040 (PI/PD Mantis). This content is solely the responsibility of the authors and does not necessarily represent the official views of the NIH.”

5. We note that Figure 4 in your submission contain copyrighted images. All PLOS content is published under the Creative Commons Attribution License (CC BY 4.0), which means that the manuscript, images, and Supporting Information files will be freely available online, and any third party is permitted to access, download, copy, distribute, and use these materials in any way, even commercially, with proper attribution. For more information, see our copyright guidelines: http://journals.plos.org/plosone/s/licenses-and-copyright.

a. You may seek permission from the original copyright holder of Figure 4 to publish the content specifically under the CC BY 4.0 license.

Reviewers' comments:

Reviewer's Responses to Questions

**Comments to the Author**

1. Is the manuscript technically sound, and do the data support the conclusions?

Reviewer #1: Yes

Reviewer #2: Yes

2. Has the statistical analysis been performed appropriately and rigorously?

Reviewer #1: Yes

Reviewer #2: Yes

3. Have the authors made all data underlying the findings in their manuscript fully available?

Reviewer #1: Yes

Reviewer #2: Yes

4. Is the manuscript presented in an intelligible fashion and written in standard English?

Reviewer #1: Yes

Reviewer #2: Yes

Reviewer #1: This report by Palmer, et al., describes the role of the OspA LA-2 targeting antibodies in protection from B. burgdorferi infection through studies of an Fc-silent LA-2 monoclonal antibody/ The report is thorough and the experiments are well-designed. Some of the results were difficult to follow because of incomplete description of methods. Overall, this is an interesting study with moderate importance for the understanding of OspA subunit Lyme disease vaccine efficacy.

Methods:

-MIA should be spelled out and the recombinant proteins should be listed in that context, with a description that the assay is used to determine seroconversion in mice.

-For the mouse model of intradermal Bb challenge, please include references on the susceptibility of Balb/c mice (not typically used for Lyme disease studies) and on the expression of OspA from spirochetes grown in culture.

Results:

-define gMFI

-describe in the complement-mediated Borreliacidal assay how the MFI level relates to live Borrelia (i.e. that IPTG induction after the treatment would produce a signal if Borrelia were not killed). This will aid interpretation of Figure 3.

-With the tick-mediated challenge, why were only bladders collected, whereas in the dermal challenge, spleens, skin and joints were collected?

-Does Table 3 only reflect seroconversion? This should be included in the legend.

Line 402 subtitle is poorly written (i.e. “the B. burgdorferi skin”).

-For the experiment described in Figure 4, be clear about the # of mice per group (it looks like 2-4) and that the mice were euthanized for each biopsy (i.e. that multiple time points were not collected from each mouse). Also, for this experiment, lines 410-411 indicate that mice had positive cultures from internal tissues. If tissues of the treated mice were positive or negative, clearly state or show these data.

The legend for Fig. 5 needs to be re-worded, and it would be “intradermally infected.” As written, the treatment appears to be at 5 days post-injection. Also. For the results shown in Fig. 5, no “antibody without Bb” is shown. Ideally, LA-2 alone would be used, but if it was just ZAC-3, then show those results.

-Lines436-438: omit discussion of ongoing experiments if you do not plan to show the data in this report.

Discussion:

Here is a good opportunity to include the role of OspA in tick midgut binding (via TROSP-A). If the Bb remain in midgut after passive immunization with LA-2 antibody, then it doesn’t likely interfere with midgut binding.

Grammatical:

-Line 60 “identification of OspA”

-line 211 “allowed to molt”

Reviewer #2: This paper by Palmer and Mantis is very well written and it is an important contribution to the field. They generated and characterized an Fc-silent version of the anti-OspA monoclonal antibody LA-2 to dissect complement-dependent and complement independent activity contribution to protection against Bb. They found that LA-2 main mode of action involves direct physical interaction with the spirochete rather than complement dependent killing. I recommend that the authors add the missing data on Fig 5 and add some clarification in discussion regarding LA-2 protection in the ID model due to cultured Bb expressing OspA which is opposite to OspA being downregulated in the tick. This explains why the ID challenge works so beautifully. Also make sure to clarify that the ticks used carry B. burgdorferi sensu stricto B31. This paper was a joy to read.

Abstract

Add B. burgdorferi sensu stricto B31 throughout the manuscript in text and figure captions.

Results

Bb downregulates OspA expression in the tick midgut before transmission to the host but cultured Bb express OspA on the surface. For the mouse model of ID dissemination, please link your fig1 showing OspA expression on the surface of Bb B31 - that would be a representation of cultured live Bb used. Or add it to the discussion.

Line 363, check what ticks are used in ref 9, 32 and 46, is it Bb B31 or ticks carrying other strains?

Table 2. Is the result of the IC control discussed?

Line 400: “antibody is limiting” or “antibody is limited” ?

For Fig 5. Need to add the cytokine data for LA-2 LALAPG.

Discussion

Lines 470: We don’t really understand it completely but in my lab we often see at least a significant decrease in Bb in nymphal ticks infected with >10 strains of Bb (10, 13 and 19 strains) that feed on OspA vaccinated mice (an example here for an 8-fold reduction (PMID: 16198456 DOI: 10.1016/j.vaccine.2005.08.089), and sometimes Bb are completely absent from the tick; we have more recent unpublished data. The function of LA-2 is conclusively shown in this paper as true, so I guess it is important to make sure that the reader understands this is a function specific to LA-2 and ticks carrying B. burgdorferi sensu stricto B31. Adding LA-2 to the title of the paper will help with that concept, but I would leave a margin for the function of other non-LA-2 anti-OspA antibody.

In Table 2 the IC control clears Bb but this is not discussed.

**Do you want your identity to be public for this peer review?** For information about this choice, including consent withdrawal, please see our Privacy Policy

Reviewer #1: No

Reviewer #2: **Yes:** Maria Gomes-Solecki

---

## [Author Response · Author response to Decision Letter 1]

18 Feb 2026

Reviewer #1 (Comments for the Author): This report by Palmer, et al., describes the role of the OspA LA-2 targeting antibodies in protection from B. burgdorferi infection through studies of an Fc-silent LA-2 monoclonal antibody/ The report is thorough and the experiments are well-designed. Some of the results were difficult to follow because of incomplete description of methods. Overall, this is an interesting study with moderate importance for the understanding of OspA subunit Lyme disease vaccine efficacy.

Methods:

1. MIA should be spelled out and the recombinant proteins should be listed in that context, with a description that the assay is used to determine seroconversion in mice. Response: We spelled out Multiplex Immunoassay (MIA) in this portion of the methods. We also edited this section to make it clearer that this assay is used to determine seroconversion in mice, including adding an opening overview sentence describing the purpose of this method.

2. For the mouse model of intradermal Bb challenge, please include references on the susceptibility of Balb/c mice (not typically used for Lyme disease studies) and on the expression of OspA from spirochetes grown in culture. Response: An additional reference was added describing the susceptibility of BALB/c mice to B. burgdorferi infection and their viability in intradermal studies at the beginning of this section of the methods (Ma et al., 1997; doi: 10.1128/iai.66.1.161-168.1998/). To emphasize that the ID challenge works because the spirochetes that we inject are nearly uniformly OspA positive in vitro, we have referred to Figure 1 in the results section “LA-2 and LA-2 LALAPG passively prevent mice from intradermal B. burgdorferi s.s. B31 infection”, highlighting that the same strain was used in ID challenge experiments. We also added a reference (Srivastava, 2008; DOI: 10.1128/JB.00085-08) which supports nearly uniform OspA expression on B. burgdorferi s.s. B31 spirochetes cultured in vitro.

Results:

1. Define gMFI Response: We spelled out this term in the Figure 1 legend.

2. Describe in the complement-mediated Borreliacidal assay how the MFI level relates to live Borrelia (i.e. that IPTG induction after the treatment would produce a signal if Borrelia were not killed). This will aid interpretation of Figure 3. Response: We added a sentence to this portion of the results clarifying that if cells are not killed, we expect to see a fluorescence signal due to IPTG induction, and that this signal is quantitative. Additionally, we removed the “data not shown” pieces of this result to better narrow the focus on Figure 3.

3. With the tick-mediated challenge, why were only bladders collected, whereas in the dermal challenge, spleens, skin and joints were collected? Response: For the short-term intradermal challenge (Fig 4), we collected spleens, knee joints, and skin biopsies to better understand Borrelia dissemination kinetics in the presence/absence of LA-2 (LALA-PG); we have since updated the methods and results text for the short-term intradermal study to better clarify why we collected these specific tissues. For the tick-challenge, we were focusing on whether mice were infected or not when treated with mAb, so serology was the main readout; but we also happened to have bladder culture data for this experiment. We decided to include the bladder culture data to confirm infection status in these tick-challenged mice; the methods text for the tick challenge has been updated to confirm why bladders were also collected in addition to serum.

4. Does Table 3 only reflect seroconversion? This should be included in the legend. Response: Correct. We have updated the legend of Table 3 to clarify this.

5. Line 402 subtitle is poorly written (i.e. “the B. burgdorferi skin”). Response: Corrected, should have read B. burgdorferi-infected mouse skin.

6. For the experiment described in Figure 4, be clear about the # of mice per group (it looks like 2-4) and that the mice were euthanized for each biopsy (i.e. that multiple time points were not collected from each mouse). Also, for this experiment, lines 410-411 indicate that mice had positive cultures from internal tissues. If tissues of the treated mice were positive or negative, clearly state or show these data.

Response: In the results text describing this experiment, we changed the language to make it clearer about the number of mice per group. On the second point, we have since clarified that it was only the untreated groups in which distal tissues were positive, not mAb treated.

7. The legend for Fig. 5 needs to be re-worded, and it would be “intradermally infected.” As written, the treatment appears to be at 5 days post-injection. Also. For the results shown in Fig. 5, no “antibody without Bb” is shown. Ideally, LA-2 alone would be used, but if it was just ZAC-3, then show those results.

Response: As requested, we have corrected the Figure 5 legend to better reflect the experiment.

As for the condition of “antibody without Bb”, this is not something we tested for this experiment, as we felt confident that circulating mAb would not influence local inflammation of the ID injection site. We administer mAb subcutaneously at the nape of the neck, whereas we ID infect mice on the right or left flank, so local inflammation caused by the SC mAb injection process should not be an issue of concern. We were concerned if inflammation caused by the ID process would obscure the result when Bb was introduced, however we addressed this by having an uninfected group.

8. Lines 436-438: omit discussion of ongoing experiments if you do not plan to show the data in this report. Response: We removed the sentence describing ongoing experiments from the text.

Discussion:

1. Here is a good opportunity to include the role of OspA in tick midgut binding (via TROSP-A). If the Bb remain in midgut after passive immunization with LA-2 antibody, then it doesn’t likely interfere with midgut binding. Response: We appreciate the reviewer’s suggestion and overlooked mentioning the role of TROSPA in the original submission. We have since added this text to the discussion:

“However, other mechanisms may also be at play. Gipson and de Silva, for example, speculated antibodies like C3.78 influence spirochete gene expression patterns, including the requisite ospA to ospC transition during transmission{Gipson, 2005 #6596;Wang, 2016 #5726}. It is also possible that passive immunization with LA-2 and LA-2 LALAPG antibody do not result in a decline in tick midgut burdens because they do not interfere with OspA binding to tick receptor for OspA (TROSPA) in the tick midgut. TROSPA is highly expressed in flat nymphs and downregulated upon engorgement, serving as an anchor of B. burgdorferi s.s. B31 to its vector {Pal, 2000 #7234;Pal, 2004 #7412} Whether competition between the two occurs has yet to be elucidated.”

Grammatical:

1. Line 60 “identification of OspA” Response: Corrected

2. line 211 “allowed to molt” Response: Corrected

Reviewer #2 (Comments for the Author): This paper by Palmer and Mantis is very well written and it is an important contribution to the field. They generated and characterized an Fc-silent version of the anti-OspA monoclonal antibody LA-2 to dissect complement-dependent and complement independent activity contribution to protection against Bb. They found that LA-2 main mode of action involves direct physical interaction with the spirochete rather than complement dependent killing. I recommend that the authors add the missing data on Fig 5 and add some clarification in discussion regarding LA-2 protection in the ID model due to cultured Bb expressing OspA which is opposite to OspA being downregulated in the tick. This explains why the ID challenge works so beautifully. Also make sure to clarify that the ticks used carry B. burgdorferi sensu stricto B31. This paper was a joy to read.

Abstract

1. Add B. burgdorferi sensu stricto B31 throughout the manuscript in text and figure captions. Response: In instances where we used the B. burgdorferi sensu stricto B31 strain, we have incorporated this into the text (spelled out first and subsequently referred to as B. burgdorferi s.s. B31).

Results

1. Bb downregulates OspA expression in the tick midgut before transmission to the host but cultured Bb express OspA on the surface. For the mouse model of ID dissemination, please link your fig1 showing OspA expression on the surface of Bb B31 - that would be a representation of cultured live Bb used. Or add it to the Discussion. Response: We appreciate this suggestion from the reviewer. To emphasize that the ID challenge works because the spirochetes that we inject are nearly uniformly OspA positive in vitro, we have referred to figure 1 in the results section “LA-2 and LA-2 LALAPG passively prevent mice from intradermal B. burgdorferi s.s. B31 infection”, highlighting that the same strain was used in ID challenge experiments. We also added a reference (Srivastava, 2008; DOI: 10.1128/JB.00085-08) which supports nearly uniform OspA expression on B. burgdorferi s.s. B31 spirochetes cultured in vitro.

2. Line 363, check what ticks are used in ref 9, 32 and 46, is it Bb B31 or ticks carrying other strains? Response: This is a good point that we did not consider. After looking again, ticks in references 9 and 46 harbor B. burgdorferi s.s. N40, and ticks in reference 32 harbor the Westchester strain. Looking back, it also appears that in the case of mAb C3.78, in which a low dose (~60 ug/mouse) antibody protected mice from tick-mediated B. burgdorferi challenge without a concomitant reduction in spirochete numbers in tick tissues (ref 47), as mentioned in the discussion and matching what we saw with LA-2, the ticks harbored B31. Perhaps the strain used is more pertinent than we initially thought? At this line, we noted that in references 9, 32 and 46 that alternate strains were used, whereas B. burgdorferi s.s. B31 was used in the case of C3.78.

3. Table 2. Is the result of the IC control discussed? Response: We added the IC result to the results text of that; also please refer to the response to discussion question 2 for more information on the IC groups in the tick challenge experiment

4. Line 400: “antibody is limiting” or “antibody is limited”? Response: Changed to limited for clarity.

5. For Fig 5. Need to add the cytokine data for LA-2 LALAPG. Response: Unfortunately, we do not have this data at this time. If necessary, we can redo this experiment and incorporate LA-2 LALAPG. Because the CBA analysis is not our main focus of the manuscript, we thought it was appropriate to examine the influence of LA-2 only on cytokine influx in the skin after B. burgdorferi infection, as this is still a novel finding.

Discussion

1. Lines 470: We don’t really understand it completely but in my lab we often see at least a significant decrease in Bb in nymphal ticks infected with >10 strains of Bb (10, 13 and 19 strains) that feed on OspA vaccinated mice (an example here for an 8-fold reduction (PMID: 16198456 DOI: 10.1016/j.vaccine.2005.08.089), and sometimes Bb are completely absent from the tick; we have more recent unpublished data. The function of LA-2 is conclusively shown in this paper as true, so I guess it is important to make sure that the reader understands this is a function specific to LA-2 and ticks carrying B. burgdorferi sensu stricto B31. Adding LA-2 to the title of the paper will help with that concept, but I would leave a margin for the function of other non-LA-2 anti-OspA antibody. Response: The reviewer raises an interesting point and further highlights that vaccination with OspA can lead to a reduction in tick/midgut spirochete burdens; possibly suggesting a killing mechanism. Given that with mAb C3.78 there is not a concomitant reduction in spirochete numbers in tick tissues (ref 47), as mentioned in the discussion, and this matches the LA-2 phenotype, we speculate if there is something about a polyclonal response vs monoclonal treatment which gives rise to visible reduction in spirochete numbers in the tick. However, to avoid this speculation in the manuscript, we added a line in the aforementioned paragraph (line 470) that qualifies the result such that it not be interpreted to expand beyond these two mAbs and ticks carrying B. burgdorferi s.s. B31. We have also added the reviewer’s reference above in the manuscript to provide additional evidence that historically, vaccination with OspA has coincided with a reduction/elimination of spirochetes from infected ticks used in challenge.

2. In Table 2 the IC control clears Bb but this is not discussed. Response: Our apologies for not making this clearer. There are actually two IC groups in table 2, one received infected ticks, and the other received naïve ticks. Infection status of each group’s ticks is seen in the 3rd column from the left; however, we added a note to the table caption to better clarify this for the reader. The IC group challenged with infected ticks was completely infected, whereas the IC group challenged with naïve ticks did not, as expected.

---

## [Editor Report · Decision Letter 1]

25 Feb 2026

An Fc-silent OspA monoclonal antibody passively protects mice from tick and intradermal Borrelia burgdorferi challenge

PONE-D-25-66161R1

Dear Dr. Mantis,

We’re pleased to inform you that your manuscript has been judged scientifically suitable for publication and will be formally accepted for publication once it meets all outstanding technical requirements.

Kind regards,

Brian Stevenson, Ph.D.

Academic Editor

PLOS One
---

## [Editor Report · Acceptance letter]

PONE-D-25-66161R1

PLOS One

Dear Dr. Mantis,

I'm pleased to inform you that your manuscript has been deemed suitable for publication in PLOS One. Congratulations! Your manuscript is now being handed over to our production team.

Kind regards,

on behalf of

Prof. Brian Stevenson

Academic Editor

PLOS One